# Spatial–Temporal Variations in Atmospheric Factors Contribute to SARS-CoV-2 Outbreak

**DOI:** 10.3390/v12060588

**Published:** 2020-05-27

**Authors:** Raffaele Fronza, Marina Lusic, Manfred Schmidt, Bojana Lucic

**Affiliations:** 1Biocomputing Unit, Genewerk GmbH, 69120 Heidelberg, Germany; raffaele.fronza@genewerk.de (R.F.); manfred.schmidt@genewerk.de (M.S.); 2Department of Infectious Diseases, Integrative Virology, Heidelberg University Hospital, 69120 Heidelberg, Germany; marina.lusic@med.uni-heidelberg.de; 3German Center for Infection Research, Partner Site Heidelberg, 69120 Heidelberg, Germany

**Keywords:** SARS-CoV-2, infection dynamics, viral outbreak, PM_2.5_, ozone

## Abstract

The global outbreak of severe acute respiratory syndrome coronavirus 2 (SARS-CoV-2) infection causing coronavirus disease 2019 (COVID-19) has reached over five million confirmed cases worldwide, and numbers are still growing at a fast rate. Despite the wide outbreak of the infection, a remarkable asymmetry is observed in the number of cases and in the distribution of the severity of the COVID-19 symptoms in patients with respect to the countries/regions. In the early stages of a new pathogen outbreak, it is critical to understand the dynamics of the infection transmission, in order to follow contagion over time and project the epidemiological situation in the near future. While it is possible to reason that observed variation in the number and severity of cases stems from the initial number of infected individuals, the difference in the testing policies and social aspects of community transmissions, the factors that could explain high discrepancy in areas with a similar level of healthcare still remain unknown. Here, we introduce a binary classifier based on an artificial neural network that can help in explaining those differences and that can be used to support the design of containment policies. We found that SARS-CoV-2 infection frequency positively correlates with particulate air pollutants, and specifically with particulate matter 2.5 (PM_2.5_), while ozone gas is oppositely related with the number of infected individuals. We propose that atmospheric air pollutants could thus serve as surrogate markers to complement the infection outbreak anticipation.

## 1. Introduction

The pandemic of the new coronavirus SARS-CoV-2 causing COVID-19 disease is testing the resilience of all the vital components of our communities, and specifically healthcare systems. The majority of new infections are now being reported outside of China, where the outbreak officially originated in December 2019 in Wuhan.

SARS-CoV-2 belongs to a family of Coronaviridae, the common spillover pathogens that have recently caused two viral outbreaks, SARS and the Middle East respiratory syndrome (MERS). As SARS-CoV and MERS-CoV, new SARS-CoV-2 most probably originated from bats, however the intermediate host of the new coronavirus remains unknown to date [1]. Like other coronaviruses, SARS-CoV-2 is a spherical enveloped virus with the positive-sense RNA genome. In general, coronaviruses have a relatively slow viral multiplication rate with the in vitro maximal replication efficiency at 32–33 °C, and a rapid decrease in infectivity at higher temperatures [2]. Moreover, coronaviruses can remain infectious for several days on inert surfaces and in the external environment [3,4]. 

The general indicator of transmissibility of a viral infection, basic reproduction number (R_0_), which indicates how many secondary infections are caused by a primary infected person, has been estimated for SARS-CoV-2 to be between 2 and 3, mostly using data from China [5]. However, these estimations might not be accurate for what seems to be the faster-spreading European epidemic. The main transmission mode of SARS-CoV-2 is person-to-person contact [6]. Besides physical contact, transmission can occur through physiological aerosol airborne droplets composed mainly of water in the act of breathing, talking, coughing and sneezing [7,8]. The droplets that compose aerosol include various physiological constituents (cells, proteins, salts, small molecules) but also contain exogenous particles, such as viruses [7]. The dimension of these droplets varies, ranging from millimeters to microns, leading to a substantial difference in the detrimental effect in the case of infective particles [9,10]. Large droplets have a low probability of passing the upper respiratory tract, whereas the smaller particles have the capacity to reach the bronchi and lungs [7]. It is considered that droplets independent from the dimension, fall promptly to the ground as subordinated to gravity force, explaining the reason for the heuristic rule of maintaining a distance of a few meters to avoid the infection. However, the gravity force is counteracted by Stokes friction and, in the case of particles up to 4 μm, the two contrasting forces are equilibrating and the particles are floating in the air [11]. 

Air pollutants PM_2.5_ and PM_10_ are atmospheric aerosols that are classified on the basis of the aerodynamic particulate size (i.e., <2.5 and <10 μm). In-depth analysis and re-analysis of large cohort studies around the world showed evidence that both short- and long-term exposure to particulate matter is associated with all-cause and cardiopulmonary mortality (reviewed in [12,13]). General hypothesized pathophysiological mechanisms that link PM exposure to cardiopulmonary diseases involve the rapid progression of existing pulmonary conditions, pulmonary and systemic oxidative stress, inflammation, atherosclerosis and accompanying cardiovascular diseases [14,15]. This is supported by clinical studies showing that air pollution, and specifically particulate matter PM_2.5_, positively correlates with the number of outpatient and emergency visits and hospitalizations for acute respiratory infections [16,17]. Toxicological studies suggest that PM_2.5_ are especially harmful as smaller particles are more prone to penetrate deeper into the lungs [18,19]. Strikingly, ultrafine particles (<100 nanometres in diameter) have been also found in the brain, indicating that PM effects are not limited to respiratory and cardiovascular systems [20,21]. Similarly to particulate matter, atmospheric pollutant ozone (O_3_), an oxidant present in the environment, also contributes to the risk of respiratory illness [22]. Furthermore, both PM_2.5_ and O_3_ are found among the most prominent environmental cardiovascular risk factors [23]. However, unlike particulate matter, ozone is commonly used as a method for air, water and object disinfection. Indeed, in the case of an airborne influenza virus type A infection, O_3_ was found to rapidly inactivate the virus and to reduce morbidity in infected mice [24,25,26]. 

It is generally recognized that respiratory viruses, such as influenza virus and SARS-CoV, can be transmitted by airborne diffusion of infectious droplets generated by coughing or sneezing via the nuclei of small aerodynamic diameter [27]. We thus hypothesized that airborne transmission of SARS-CoV−2 can be influenced by, but is not limited to, indirect action of certain atmospheric conditions that maintain infectious nuclei suspended for prolonged periods of time, parameters that also act on atmospheric pollutants. In this study, we analyzed particulate matter and ozone ambiental concentration in relation to SARS-CoV-2 infected cases from Italy and three other European countries: France, Germany and Spain. We propose that the same atmospheric conditions that elevate the air concentration of PM and O_3_ are also contributing to the modulation of the viral outbreak. We provide a qualitative model that predicts the outbreak severity taking into account only a reduced set of atmospheric factors. We propose that (1) atmospheric conditions that elevate the PM concentration are also involved in the SARS-CoV-2 viral outbreak, and that (2) the environmental ozone concentration can be a repressing factor that attenuates SARS-CoV-2 infection.

## 2. Data and Methods

### 2.1. Study Setting and Data

The hourly concentration (μg/m^3^) of PM_2.5_, PM_10_, O_3_ and NH_3_ in 47 regional capitals of Europe, and 107 major Italian cities, was retrieved in the period from 10th of February 2020 to 10th of April 2020. For the same period, we obtained the hourly temperature T (°K), the dewpoint temperature TD (°K), the u and v speed components of the wind (m s^−1^), and the surface pressure P (Pa). For seasonal analysis, the first 30 days of March, June, September, and December were taken from the year 2018. The files are in crib2 or netCDF format retrieved from http://macc-raq-op.meteo.fr/. The concentration of pollutants PM_2.5_, PM_10_, O_3_ and NH_3_ were extracted from the ENSEMBLE multi-model that combines the values of other seven models: CHIMERE METNorway, EMEP RIUUK, EURADIM KNMI/TNO, LOTOS-EUROS SMHI, MATCH FMI, SILAM Météo-France, and MOCAGE UKMET (https://www.regional.atmosphere.copernicus.eu). The climate factors T, TD, wind components and P are obtained from the Copernicus Climate Change Service Climate Data Store CDS [28]. The European domain is defined within the coordinates −25E/70N and 45E/30N, with 0.1 and 0.28125 degrees of horizontal and vertical resolution, respectively. The values were extracted and processed into a Linux environment using bash scripts, cdo 1.7.0 and the R packages raster and ncdf4 [29,30,31]. 

The geographical center of the province capital corresponds to the geometric center of the quadrant. The latitude and longitude of Italian cities were derived directly from the epidemiological data provided by the Protezione Civile. The latitude and longitude of European cities were obtained manually. 

Italian province dataset was subdivided in the three macroregions: north Italian provinces (latitude ≥ 44.85), central Italian provinces (41.5 ≤ latitude ≥ 44.85), and south Italian provinces (latitude ≤ 41.5).

To express the overall quantity in the study period for PM_2.5_, PM_10_, NH_3_, and O_3_, we developed and tested two indexes, the average daily maximum (AdM) and the average daily value (AdV). Considering, as an example, the starting day of the atmospheric factors sampling (22nd of February 2020), the incubation period (6 days, [32]) and the day at which we started the survey (25th of March 2020), D was imposed as D=27. Considering that there are *L* geographical areas, and 24 h a day (*H*), the AdM at a day *d* is a vector computed as:$$AdM_{l}=\frac{1}{D}\overset{D}{\underset{d=1}{\sum }}max_{h}\left(x_{hdl}\right)\;$$
where $x_{hdc}$ is the atmospheric air pollution value at hour *h*, day *d*, and location *l*. The AdV is computed as:$$AdV_{l}=\frac{1}{DH}\overset{D}{\underset{d=1}{\sum }}\overset{H}{\underset{h=1}{\sum }}x_{hdl}$$

The two indexes are similar with AdM being more sensitive to rapid changes of the factors whereas AdV is more robust to spikes and responses to temporal consistent changes. For ozone, the concentrations were converted from μg/m^3^ to ppm assuming standard conditions ($273.1^{{}^{\circ}}K,\;1$ atm) and using the relation:$$ppm=c\cdot \frac{MV}{MM}\cdot 10^{-3}=0.467\;10^{-3}\cdot c$$
where *c* is the ozone concentration $\left(\mu g/m^{3}\right)$, *MV* is the molar volume approximated with an ideal gas (22.414 L/mol) and *MM* is the molar mass (48 g/mol).

The average daily temperatures, *AdTD* and *AdT*, and the average daily surface pressure *AdP* are obtained similarly to *AdV*
$$AdTD_{l}=\frac{1}{DH}\overset{D}{\underset{d=1}{\sum }}\overset{H}{\underset{h=1}{\sum }}td_{hdl}$$
$$AdT_{l}=\frac{1}{DH}\overset{D}{\underset{d=1}{\sum }}\overset{H}{\underset{h=1}{\sum }}t_{hdl}$$
$$AdP_{l}=\frac{1}{DH}\overset{D}{\underset{d=1}{\sum }}\overset{H}{\underset{h=1}{\sum }}p_{hdl}$$
where *t_hdl_*, *td_hdl_*, and *p_hdl_* are the temperature, the dewpoint temperature, and the surface pressure at hour *h*, day *d*, and location *l* respectively.

The average daily relative humidity *AdRH* is obtained by applying the Magnus approximation formula [33]:$$AdRH_{l}=100\;e^{\frac{4283.58\left(AdTD_{l}-AdT_{l}\right)}{\left(243.04+AdTD_{l}\right)\left(243.04+AdT_{l}\right)}}$$

The magnitude of wind vector *AdW* is calculated as
$$AdW_{l}=\frac{1}{DH}\overset{D}{\underset{d=1}{\sum }}\overset{H}{\underset{h=1}{\sum }}\sqrt{u_{hdl}^{2}+v_{hdl}^{2}}$$
where *u_hdl_* and *v_hdl_* are the west-east and south-north components of the wind at hour *h*, day *d*, and location *l* respectively.

For each species, we obtained a 1xn vector at a last day *D*, where *n* is the number of areas considered. 

### 2.2. Epidemiological Data

Given the open data policy of the Italian Government, for Italy it was possible to obtain the daily data of the SARS-CoV-2 infected people per province and region. The demographic data were collected from the site http://www.comuni-italiani.it/provincep.html. The Italian SARS-CoV-2 case data are retrieved from https://github.com/pcm-dpc/COVID-19 in the form of daily, comma-separated files. For each day, we collected a set of m×n matrices, where *m* is the number of days and *n* is the number of the provinces or regions.

In the case of the other three European countries, France, Germany and Spain (FGS), the data were not centralized or specified at the time of the analysis and were manually imported without the possibility of being separated into subgroups of cases. The total number of cases for 18 French, 16 German and 19 Spanish regions were obtained from https://www.statista.com/ referring to the 25th of March. Four French and one Spanish regions were removed from our analysis, as they were located outside the topographic European domain.

### 2.3. Statistical Analysis

The pairwise Spearman Correlation Coefficients were computed for all the atmospheric factors. Conditioning plots were then applied to graphically assess the dependence of the number of infected cases per million of one factor conditioned to different levels of another factor. 

The infection data used per province are the total number of patients on 25th of March and 15th of April, whereas for the regions we used the number of total, not hospitalized and hospitalized cases. We calculated *AdM_l_*, where *l* = PM_2.5_, PM_10_, O_3_ and NH_3_ considering *D* = 27 (endpoint 19th March and 9th April). 

### 2.4. Binary Classifier

A binary classifier based on an artificial neural network (ANN) was implemented to test the capacity of the atmospheric variables to predict the epidemic escalation of the number of positive cases per million on the basis of a combination of *AdM_l_* where *l* = PM_2.5_, PM_10_, NH_3_ and O_3_. All the *AdM_l_* values were pre-scaled to be homogeneous. We defined epidemic escalation, at the testing time (25th of March and 15th of April), as incidences where the number of cases per million (at the time of the present analysis) is more than one standard deviation (σ_1_ = 480, σ_2_ = 1002), above the mean (μ_1_ = 543, μ_2_ = 1551) of the number of cases per million in the first twenty nations listed in the coronavirus survey https://www.worldometers.info/coronavirus/ and https://github.com/owid/covid-19-data/tree/master/public/data. We call this value escalation threshold (*T*_*e*1_ = 1023 cases per million and *T*_*e*2_ = 2553 cases per million at 25th of March and 15th of April, respectively). The binary classifier was developed using a feed-forward ANN with *n* inputs, one output and h hidden components using the *neuralnet* R package. The number of inputs *n* depends on the number of regressors used as input, whereas the number of the hidden components was selected during the training phase with the Italian province datasets. The dimensionality of the input parameters was finally reduced to the usage of two variables, PM_2.5_ and O_3_. 

To validate the models, we selected a Monte Carlo cross-validation strategy. The observed number of data was split randomly multiple times (*n* = 100) into two datasets, the training (*R*) and the test (*T*), containing 75 and 32 provinces, respectively. For each combination of regressors, we varied the number of hidden layers from 3 to 15. We defined as true negative (*TN*) all the provinces in *T* that have a number of cases that is lower than the selected threshold. The true positives (*TP*) are provinces where the number of cases is higher than the threshold. The false positives (*FP*) are all the provinces predicted with epidemic escalation but where the number of cases is lower than the threshold. The false negatives (*FN*) are all the provinces predicted without epidemic escalation but where the number of cases is higher than the thresholds. The random assignment to *R* or *T* was performed 100 times for each combination of the parameters. A null predictor was implemented with a random swapping of the number of cases. To measure the performances of the binary classifier and the null predictor, we used four indices: sensitivity ($SE=\frac{TP}{TP+FN}$), specificity ($Sp=\frac{TN}{TN+FP}$), accuracy ($ACC=\frac{TP+TN}{TP+FN+TN+FP}$), and precision ($PRC=\frac{TP}{TP+FP}$) for PM_2.5_, PM_10_, NH_3_, O_3_, and the combinations of the four factors. The best model and threshold that maximize the sum of all four indices on an average of 100 training–testing phases were selected. To associate the prediction results to the country maps we recorded for each administrative area, the number of times that the conditions in that province were predicted escalated when the province was assigned to the test dataset (*T*).

### 2.5. Classifier Evaluation

To evaluate the capacity of the classifier we collected the total number of SARS-CoV-2 cases for three European nations, France, Germany and Spain, from 25th of March. The atmospheric factors PM_2.5_ and O_3_ were extracted in the same way as described in the study setting and data section. The data were then scaled following the same procedure, and the conditioned factors were used to feed the classifier to measure the performances on completely unseen data. The expected number of infected cases in the total of 107 Italian provinces were predicted for the months of March (Spring), June (Summer), September (Autumn) and December (Winter) using the real measured values for PM_2.5_ and O_3_ atmospheric factors from 2018 seasonal datasets.

### 2.6. Informal Boolean Approximation

The results of the predictions and the previous results were condensed into a simplified Boolean model. The explaining continuous factors, PM_2.5_ (V(p)), and ozone (V(o)), where p and o are concentrations, are seen as boolean variables that refer to two states, present ($V(p>T_{p}),\;V(o>T_{o})$ or not present, depending on the unknown thresholds ($T_{p}$, $T_{o}$). As the atmospheric factors considered are limited source that influences the severity of infection, all the hidden factors are summarized by a single hidden variable V(u) with unknown characteristics. The effect of the combination of the three variables on the epidemic has two states, escalating and non-escalating, on the basis of the described epidemic escalation threshold $T_{e}$. This model is qualitative, and the activation thresholds $T_{p,\;o}$ are not explicitly calculated. The observation that the modality of the interactions among factors derived from the regression step is not known, justified the usage of an ANN as a general approximator.

### 2.7. Administrative Maps

The administrative maps for Italy, France, Germany and Spain were obtained from GADM service (https://gadm.org/about.html) version 3.6. The four country maps are then coloured accordingly with the number of reported cases until 25th of March and the corresponding ANN prediction. The AdM values for PM_2.5_ and O_3_ are mapped for Italy.

## 3. Results

The concentration of four air pollutants and five atmospheric variables in 107 Italian and 47 FGS areas were sampled over a period of 27 days from 00 AM on the 10th of February until the 00 AM of the 10th of April. The average population of an Italian province is 572,646 inhabitants, with an average number of cases per million of 1226 at the time of the study. On the same day, the average population and number of cases per million in the FGS regions was 4,241,214 and 1097, respectively. In the first 20 countries, ranked by the total number of cases at the day of the study, the average number of cases per million was 480 (25th of March) and 1551 (15th of April).

To measure the strength of association between all the variables in the provinces, we used the Spearman coefficients (SCs) (Appendix A). All the $AdM_{PM2.5}$, $AdM_{PM10}$, and $AdM_{NH_{3}}$ values were nearly identical with the respective $AdV_{l}\;$ values (*r* > 0.99), suggesting that the two indexes are redundant. The $AdM_{O_{3}}$ and $AdV_{O_{3}}$ values were correlated with slightly lower SC (*r* = 0.91). Correlation matrix for PM_2.5_, PM_10_ and NH_3_ shows a strong positive inter-variables correlation (*r* > 0.7). The number of SARS-CoV-2 cases per million (Cas, Table 1) shows a significant positive correlation with three of the four pollutant variables PM_2.5_, PM_10_, and NH_3_ (0.58≤r≤0.68). O_3_, instead, shows a good negative correlation with Cas (*r* = −0.44). All the atmospheric variables, except the surface pressure, are mildly inversely associated with the number of cases (−0.327 < *r* < 0.253). We introduced the density population (inhabitants/km^2^) of the province (PD) to assess a correlation between the number of inhabitants and atmospheric pollution (Table 1). A limited correlation was found between the PD, PM_2.5_ and PM_10_ (0.35426 =< *r* <= 0.3771). Moreover, no correlation was found among the density and NH_3._ For O_3,_ a modest negative correlation with PD (*r* = −0.244) was detected. For all the atmospheric variables, no association was found. The association between the pollutant and atmospheric variables is shown in Table 2. Ozone is positively associated with all the atmospheric variables except for the pressure P. Conversely, PM_2.5_, PM_10_, and NH_3_ are negatively correlated with all the atmospheric variables except for a limited positive association with atmospheric pressure. Overall, all tested pollutant variables display an extremely high degree of intra-variable correlation, allowing us to reduce the dimensionality of the variable space. High inter-variable correlation among PM_2.5_, PM_10_, and NH_3_ makes these variables redundant. 

To visually assess geographical distribution of PM_2.5_, PM_10_ and O_3_ concentration with respect to the number of SARS-CoV-2 cases, we generated scatter plot diagrams of three Italian macroregions: north, central and south (Figure 1A–C). While in the south of Italy (green dots, lat < 41.5), PM occupy the left part of the plot, remaining under 30 µg/m^3^ (PM_2.5_) and 40 µg/m^3^ (PM_10_) (Figure 1A,B), northern provinces show a more extended range of values, with the highest number of cases. The distribution of O_3_ (Figure 1C), is instead more uniform between regions, in a range of 0.03 to 0.05 ppm. When we combined the data from Italian provinces and FGS regions, the distribution of the PM (Figure 1D,E) in the FGS regions was clearly lower, similar to the value observed in the south of Italy (<30 µg/m^3^). The distribution of the ozone concentration is narrower in the FGS regions, with a shorter tail at the lower concentrations (Figure 1F).

Next, to account for relations that may be obscured by the effects of other variables, we generated a collection of conditional plots (Figure 2 and Appendix A) [34]. We analyzed PM_2.5_ (Figure 2A) and the O_3_ (Figure 2B,C) as the explanatory variables for the number of cases and PM_2.5_ (Figure 2A,C) and O_3_ (Figure 2B) as conditional values. When the variables are self-conditioning (Figure 2A,B), the number of SARS-CoV-2 cases per million depends on PM_2.5_ and O_3_ on an apparent opposite ways; PM_2.5_ at concentrations higher than 30 µg/m^3^ (fifth and sixth scatter plot in Figure 2A) and O_3_ at concentrations lower than 0.04 ppm (first and second scatter plot in Figure 2B). The number of cases per million starts to be dependent on O_3_ concentration only when the PM_2.5_ are higher than about 30 µg/m^3^ (fifth and sixth scatter plot in Figure 2C). In summary, conditional plots show that the number of cases per million appears to be non-linearly linked with the O_3_ data due to a threshold effect on the ozone linked to the PM_2.5_ concentration. The same threshold effect seems to hold true for the other two analyzed variables, PM_10_ and NH_3_ (Appendix A).

The ANN classifier with three hidden neurons (Appendix A) returned the highest score in SE, SP, ACC and PRC (Figure 3A and Appendix A), so this topology was selected to classify the datasets.

The prediction for the Italian provinces using PM_2.5_, O_3_ and both variables are summarized in Appendix A and Figure 3A, left panel. The classifier with both PM_2.5_ and O_3_ has sensitivity, specificity, accuracy, and precision that are highly significant with respect to the null predictor. The usage of only PM_2.5_ or O_3_ is still significant but with a decreased capacity to predict the escalated provinces (Appendix A). To visually represent the variables used and the prediction results of the classifier, we produced spatial maps for PM_2.5_, O_3_, the prediction of the conditions for SARS-CoV-2 escalation and the actual number of SARS-CoV-2 cases (Figure 3C–F).

For the FGS data, the results show a lower capacity of the combination of PM_2.5_ and O_3_ in classifying performances (Appendix A). When only PM_2.5_ was used (Figure 3A, right panel), the predictor behaved better than the null model for specificity, accuracy and precision. The classification efficiency of O_3_ alone was sub-performing for all the four performance indexes. 

Next, we tested the classifier seasonal predictive behavior, taking the *AvM* from the 107 Italian provinces, extracted from the seasonal recordings of March (Spring), June (Summer), September (Autumn), and December (Winter) 2018 (Figure 3B). Zero outbreaks were predicted in June and September (Summer and Autumn), whereas 19/107 outbreaks were found in March (Spring) and 95/107 in December (Winter).

Taken together, the classification results show that the low concentration of the particulate is a reliable predictor for the absence of epidemic escalation. At medium-high values, the model predicts the epidemic escalation in combination with low (high epidemic escalation likelihood), or high (low epidemic escalation likelihood) concentration of ozone well. This observation makes the ozone a possible predictor only if the quantity of particulate is significantly high. For all the conditions where the PM_2.5_ is low, the unknown factors explained by unknown variables (*V_U_*) are prevailing in explaining the epidemic escalation.

Finally, we explore the hypothesis that atmospheric conditions that favor the formation of PM and possibly viral micro droplets differentially influence the severity of SARS-CoV-2 infection. By separating hospitalized and not hospitalized cases in the 21 Italian regions, we found a significant positive correlation (*R*^2^ = 0.4891, *p* = 0.0004) between the particulate matter quantities and the number of hospitalized patients (Figure 4A), whereas for infected but not hospitalized population, this was not the case (*R*^2^ = 0.01154, *p* = 0.6430). Ozone instead showed a significant negative correlation with the hospitalized population (*R*^2^ = 0.2355, *p* = 0.0257) and a not significant negative correlation (*R*^2^ = 0.1590, *p* = 0.0734) with the not hospitalized population (Figure 4B).

To continuously monitor the capacity of our framework to predict the spread of the SARS-CoV-2 in the 154 areas, we regularly update the PM_2.5_ and O_3_ data and perform a real-time forecasting of the infection. The result of this analysis is provided in https://github.com/COVID19Upcome/Europe.

## 4. Discussion

In the present study, we have built a predictive model for the SARS-CoV-2 viral outbreak as a function of atmospheric pollutants, PM_2.5_, PM_10_, NH_3_ and O_3_. Our hypothesis is based on the correlation between viral outbreak and physicochemical factors that contribute to the enrichment of the atmospheric particulate. This was supported by the analysis of five additional climatic parameters, T, TD, W, HR and P, out of which the first four were found to be associated with the number of SARS-CoV-2 cases (Table 1), in line with what was reported recently [35].

Based on the Spearman coefficients and the conditional plots (Appendix A, Figure 2 and Appendix A), we chose to favor the use of the average daily maximum values of PM_2.5_ and O_3_ to build the ANN classifier for prediction of SARS-CoV-2 outbreaks. The ANN model with two input variables, PM_2.5_ and O_3_, and three hidden layers, was trained with 107 Italian provinces, which resulted in a significant predictive ability for all four considered performance values (Figure 3A and Appendix A). The same model, however, tested on FGS regions, showed a limited capacity to classify the epidemic escalation in those regions. A more in-depth analysis taking into account predictors with only a single atmospheric input variable resulted in the amelioration of the predictive capacity with only PM_2.5_ in the FGS regions. However, systematic underperformance with respect to the null model was observed when only the O_3_ parameter was used. These results allowed us to build an informal logic approximation that accounts also for the unknown causative variables (Appendix A). When the PM_2.5_ concentration is low, the capacity of the PM_2.5_ model to catch non-escalating regions (true negatives) is high, as few of them show outbreaks (*SP* > 0.9). We thus introduced the unknown causative variables to explain the rising case numbers when PM_2.5_ concentration is low (Appendix A, left branch of the model). However, when the PM_2.5_ concentration is high, the O_3_ is contributing positively to the capacity of the model in predicting the outbreaks (Figure 3A, *SE* = 0.64). This implies the existence of a variable threshold, in this case PM_2.5_ concentration, that acts on the O_3_ predictive capacity. This threshold is exemplified in the right branch of the model in Appendix A. In fact, the predictions in the FGS regions, where the outbreak conditions at the time of the analysis do not visually match the course of the infection, might be due to low values of PM_2.5_ (Figure 1D and Appendix A). On the other hand, the predicted outbreak areas in Italy clearly resemble the actual distribution of the infected population (Figure 1A and Figure 3C–F).

SARS-CoV-2 is an emerging pathogen, and our study is therefore subjected to possible limitations, primarily access to data and time constraints. Our model was validated based on publicly available data, different sampling policies and unknown fractions of untested/asymptomatic infected individuals, all of which could impact the accuracy of the collected datasets. Moreover, data access to the broader set of atmospheric factors with more powerful modeling strategies could be applied to establish explicit causal relation with the infection dynamic. Finally, the findings of our study should be seen in light of the ongoing new pandemic, which gives our analysis a short time window. However, verifying the classifier by using the two upper and lower margins (2 and 14 days lag), we observed slightly lowered predictive capability (Appendix A), suggesting that the 6 days lag used throughout the analysis is a good compromise for the incubation period. In the same line, by shortening the averaging widow for pollutants (*D* = 7, *D* = 14), we also probed ANN performance and observed that by increasing the exposure time, the prediction performance was higher (Appendix A). On this point, a comprehensive temporal synchronization based on epidemiological parameters, combined with the exploration of new additional climatic parameters and averaging schema, could improve the modelling of the airborne virus transmission dynamics.

Nevertheless, our data support the concept that the atmospheric conditions can both (1) promote the formation of persisting forms of airborne droplets charged with SARS-CoV-2 (PM_2.5_), and (2) reduce the activity of the virus (O_3_). We thus hypothesized that the increase in the concentration of PM_2.5_ may reflect the rise of infective droplets with a diameter inferior to 5 microns. We speculate that the fast evaporation of droplets emitted by talking or sneezing increases the sustained circulation of micro-droplets that carry a higher viral load, particularly in closed spaces.

In line with this, it has been shown for an airborne influenza virus that 49% of the viral particles are present in the droplets with a dimension between 1 and 4 μm. In general, droplets with a dimension bigger than 50 μm fall immediately to the ground, whereas particles with a dimension of 5 μm take more than an hour to reach the ground from a height of 3m [36]. Moreover, particles with an aerodynamic diameter smaller than 2 μm remain suspended in the air for hours or days and are more able to reach the alveolar regions in the lungs [11].

The droplet size can be modulated by the difference in vapor pressure, reducing it in a few seconds, at a rate of 5 μm/s [37]. Moreover, droplets small enough to penetrate to the lower respiratory tract can lead to an adverse disease outcome [38,39,40]. These observations are remarkably significant as they show that (1) the distribution of the dimension of the droplets changes immediately so that the diameter at the emission source (infected individual) is consistently bigger than the diameter at the destination (susceptible individual), and (2) the viral concentration in the droplet (viral particles/μL) increases, as the volume decreases quadratically with respect to the radius. This evidence suggests that in unfavorable ambiental conditions, monitored by PM_2.5_ formation, the amount of persistent droplets with large viral load reflects a significant increase in the hospitalized cases (Figure 4A). This supports the findings that face mask usage can be more beneficial in avoiding the spreading of the virus from an infected individual than to block the viral particles at the susceptible (healthy) individual [41]. With this regard, our findings are in agreement with studies focusing on the airborne transmission of SARS-CoV-2 that were published while our manuscript was in preparation. In particular, the spreading of the virus via nuclei small enough to remain suspended for prolonged periods of time is considered to be a likely mode of transmission of SARS-CoV-2 [42]. Furthermore, SARS-CoV-2 diffusion has been linked to the certain atmospheric conditions that stabilize aerosols, such as low temperature and low absolute humidity [35]. Possibly the two most coherent findings that support our hypothesis that PM can be used as an early direct or indirect signal of SARS-COV-2 air diffusion, and could thus protect human lives, and lower the global economic decline, have been recently reported [43,44]. The study by Setti et al. showed the presence of SARS-CoV-2 RNA on the particulate matter found in the focal point of the Italian epidemic. Research carried out in the U.S Harvard School of Public Health by Wu et al. found a significant association between particulate matter concentration and COVID-19 mortality rate in the U.S. Whether PM may be involved in the viral transmission only directly, or the virus can be also transmitted independently of actual association with PM ‘carriers’ when the conditions of atmospheric stability are favoured, remains to be assessed by further experimental studies.

Finally, the environmental factors that influence the particulate size/concentration, may be also responsible for the seasonal dynamics of the airborne infections. Our data are in agreement with this possibility (Figure 3B), however, a more detailed analysis is required to prove this causation for SARS-CoV-2 infection. To continue exploring seasonal and daily trends, the updated predictions in the 154 regions of our model with PM_2.5_ and O_3_ as input variables, can be found at https://github.com/COVID19Upcome/Europe.

Air pollutants unambiguously pose a hazard to human health [13]. However, the underlying pathways that link air pollutants to all-cause pathophysiological conditions, respiratory and cardiovascular morbidity and mortality still remain opaque due to the complex interplay between genetic predisposition, environmental as well as social components. This also seems to hold true for the present COVID-19 pandemic, where the host response to infection ranges from mild symptoms to severe respiratory conditions with multiple organ failure [45]. It is thus conceivable that some air pollutants, specifically particulate matter, might play a role in virus spread and pathogenicity. Our analysis supports the hypothesis that the atmospheric conditions that increase the particulate matter formation, are also contributing to the severity of the SARS-CoV-2 infection. An appealing possibility that ozone might act to counteract/sterilize viral charge is to be further investigated. Finally, monitoring spatial–temporal variations in atmospheric particulate and O_3_ could be used as an aid to estimate upcoming trends for the SARS-CoV-2 transmission impact.

## Figures and Tables

**Figure 1 viruses-12-00588-f001:**
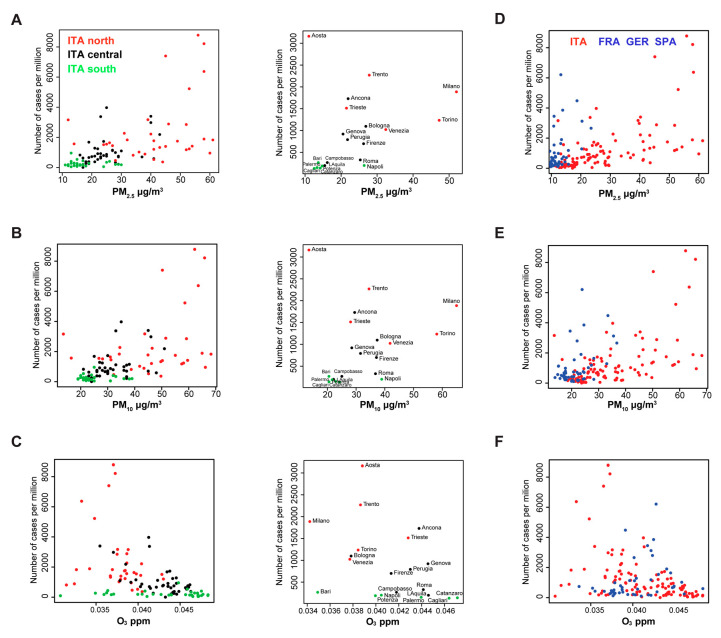
Correlation between SARS-CoV-2 cases in 107 Italian provinces using PM_2.5_, PM_10_ and O_3_. The scatterplots display the values of three atmospheric factors and the number of cases per million in 107 provinces (left panels) and selected provinces with regional capitals (middle panels). (**A**) PM_2.5_; (**B**) PM_10_; (**C**) O_3_. Different colors represent Italian provinces at different latitudes. Red dots: provinces with a latitude higher than 44.84N; black dots: provinces with a latitude comprised between 41.50N and 44.86N; green dots: provinces with a latitude lower than 41.50N. (**D**–**F**) The scatterplots display the concentration of PM_2.5_, PM_10_, and O_3_. Red dots represent Italian provinces, blue dots FGS regions.

**Figure 2 viruses-12-00588-f002:**
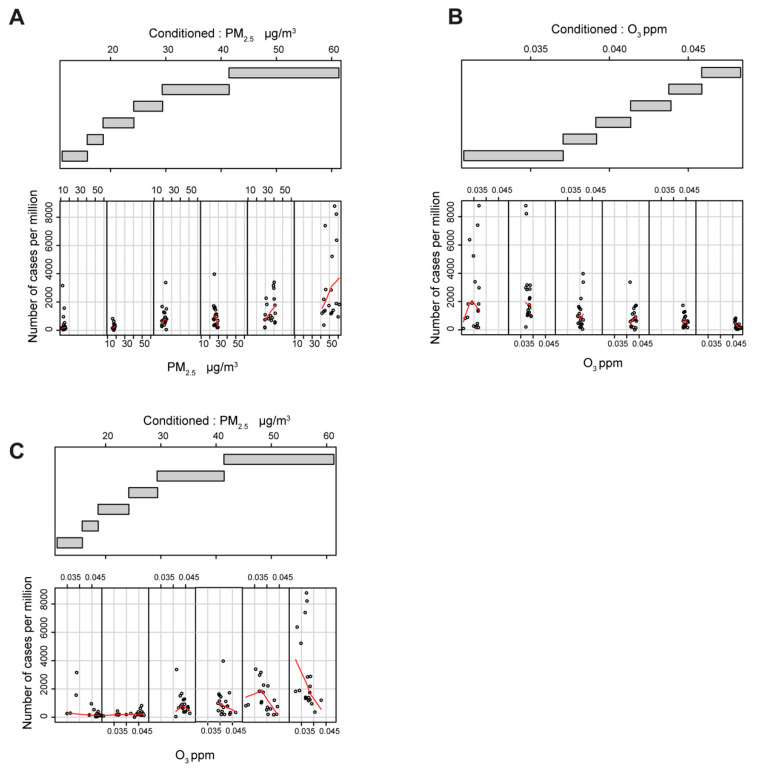
Evaluation of the cross-influence between PM_2.5_ and O_3_. Lower panel: the scatter plots between the selected factor and the number of cases per million. The LOESS curve is computed in each scatterplot and shown in red. Each plot is restricted to show datapoints that belong to the provinces that fall in the corresponding range of the conditioning factor. Upper plot: the range of the values that define each level of conditioning. The overlap among the levels is 0.1. (**A**) the scatterplot of PM_2.5_ conditioned to PM_2.5_; (**B**) the scatterplot of O_3_ conditioned to O_3_; (**C**) the scatterplot of O_3_ conditioned to PM_2.5_.

**Figure 3 viruses-12-00588-f003:**
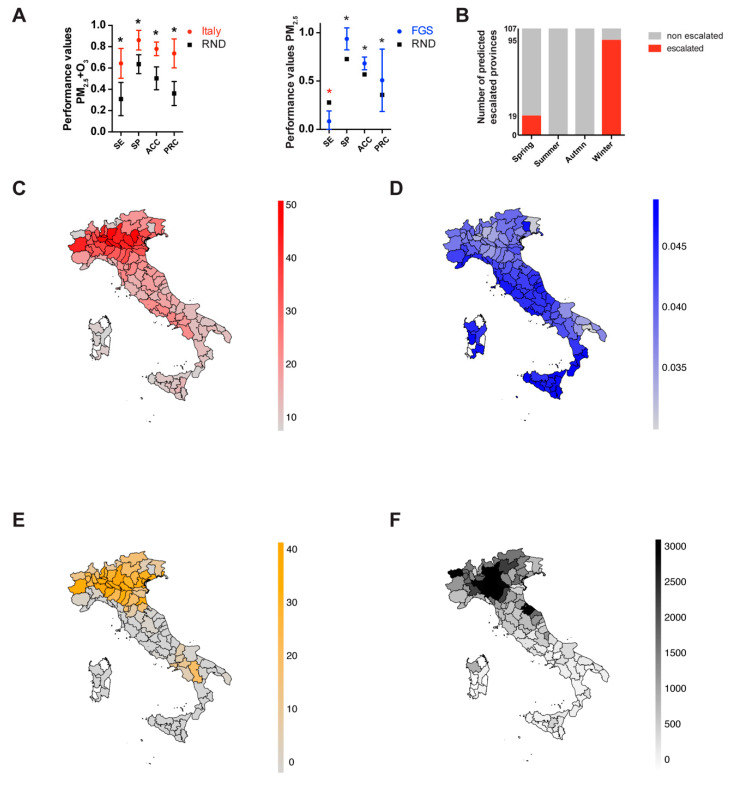
ANN performance assessment. (**A**) Performance values of the ANN on the 107 Italian provinces (left) and FGS (right) data. SE sensitivity, SP specificity, ACC accuracy, PRC precision. The dots represent the ANN average performance based on 100 Monte Carlo cross-validations. Bars represent the standard deviation. Red dots Italian provinces, blue dots FGS regions and black dots random dataset. (**B**) The histogram represents the number of the escalated Italian provinces (107) considering the PM_2.5_ and O_3_ concentrations in four months, March, June, September, and December, representative of the four seasons: Spring, Summer, Autumn, and Winter. Red bars, number of escalated provinces, grey bars, remaining non escalated provinces. Statistical analysis were performed using multiple *t* test corrected with the Sidak–Boneferroni method for multiple comparisons (*p* < 0.001). Black asterisk indicates that the classifier performs better than the null classifier. Red asterisk indicates that the classifier performs worse than the null classifier. (**C**–**F**) The spatial administrative maps representing PM_2.5_ (μg/m^3^, red), O_3_ (ppm, blue), prediction (number of positive predictions, orange) and actual reported cases (number of cases per million, black) for the Italian provinces. (**E**) The color intensity on the map represents the number of times that provinces in the test dataset were positive for the outbreak in one hundred Monte Carlo cross-validations. (**F**) The number of actual reported cases was limited to 3000 per million to increase the dynamic range of the map.

**Figure 4 viruses-12-00588-f004:**
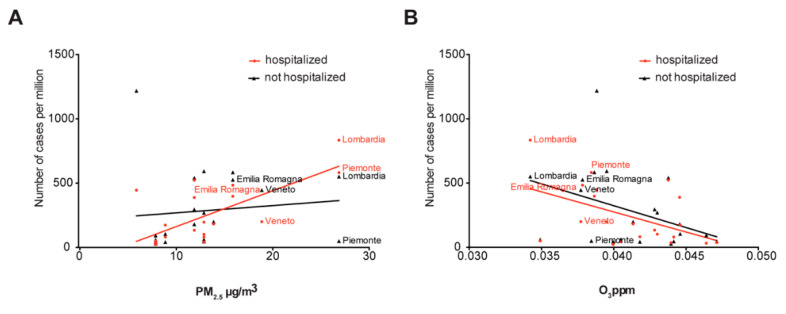
Correlation between hospitalized and not hospitalized cases in 21 Italian regions using PM_2.5_ and O_3_. (**A**) Scatter plot of the hospitalized (red points) and not hospitalized (black points) versus the concentration of PM_2.5_ in 21 Italian regions. Red line, positive correlation (*R*^2^ = 0.4891, *p* = 0.0004); black line, no significant correlation (*R*^2^ = 0.01154, *p* = 0.6430). The labels represent the four regions with the highest number of cases per million. (**B**) Scatter plot of the hospitalized (red points) and not hospitalized (black points) versus the concentration of ozone in 21 Italian regions. Red line, negative correlation (*R*^2^ = 0.2355, *p* = 0.0257); black line negative correlation (*R*^2^ = 0.1590, *p* = 0.0734). The labels represent the four regions with the highest number of cases per million.

**Table 1 viruses-12-00588-t001:** The correlation vectors between population density (PD), PM_2.5_, PM_10_, NH_3_, O_3_, dew-point temperature (TD), temperature (T), relative humidity (RH), wind (W), pressure (P) and number of SARS-CoV-2 cases in Italian provinces (Cas).

	PD	Cas
PD	1.000	0.034
PM_10_	0.377	0.586
PM_2.5_	0.354	0.597
NH_3_max	0.084	0.693
O_3_max	−0.244	−0.444
TD	0.045	−0.327
T	0.094	−0.284
RH	−0.108	−0.253
WIND	0.006	−0.327
P	0.188	−0.025

**Table 2 viruses-12-00588-t002:** The correlation matrix among the pollutant variables (PM_2.5_, PM_10_, NH_3_, O_3_) and the atmospheric variables (TD, T, RH, and P) in the Italian provinces.

	TD	T	RH	WIND	P
PM_10_	−0.209	−0.136	−0.282	−0.441	0.269
PM_2.5_	−0.300	−0.221	−0.336	−0.520	0.192
NH_3_	−0.227	−0.168	−0.253	−0.377	0.164
O_3_	0.471	0.351	0.502	0.439	0.025

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
