# Peer review of "Spatial–Temporal Variations in Atmospheric Factors Contribute to SARS-CoV-2 Outbreak"

_viruses, 2020, doi:10.3390/v12060588_

Round 1

Reviewer 1 Report

This manuscript explores the potential role that atmospheric factors (air pollution and meteorological factors) have in predicting outbreak patterns of COVID-19 in Europe. Authors run an artificial neural network and present basic statistical results, including correlations and regression output. My main concern with the paper is with the Poisson regression model used here, which in my opinion is not appropriate to model the association due to violations of independence (more on this below). My recommendation would be to remove that portion of the manuscript, and just make minor edits on the other sections. Please see my specific comments below:

Major comments:
1. Independence violation. One of the basic assumptions of linear regression is independence, that is, Yi's are independent from each other given Xi's. In a typical time-series where the outcome is non-communicable, the assumption holds. However, the critical difference for an infectious disease is, of course, that it is infectious. This is a direct violation of independence, and linear regression in this case would not be appropriate. In addition, there are a plethora of additional problems that make this modeling particularly difficult: policy decisions, worldwide travel bans, changes in consumer behavior, backlogged testing, to name just a few. Because the focus of this manuscript is not on this analysis anyways, I recommend just removing it and focus on the other analyses. I think it is appropriate to present correlations, but anything beyond that may not be appropriate.

2. Expanded discussion of health effects. Potential health mechanisms of air pollutants are mentioned very briefly. The literature of this subject is quite vast, and I'm sure there are additional toxicological studies conducted after the 1980s, some of which the authors referenced. For example, consider the landmark critical review on particulate matter by Pope and Dockery (citation below). This should also be expanded in the discussion section as well.

Minor comments:
1. Have the authors considered lag exposure? It appears only same-day exposures are considered.

2. The "2.5" and "10" after PM should be subscript.

3. Because COVID-19 related research is being outputted at such a rapid rate at the moment, there may have been significant changes/new papers on this topic just during the time of this review. Please review the existing literature and add references/adjust text accordingly.

Citation:

C. Arden Pope III & Douglas W. Dockery (2006) Health Effects of Fine Particulate Air Pollution: Lines that Connect, Journal of the Air & Waste Management Association, 56:6, 709-742, DOI: 10.1080/10473289.2006.10464485

Reviewer 2 Report

I want to congratulate authors for completing this research during COVID-19 pandemic. In this manuscript, the authors analyzed the ambient air pollution and COVID-19 cases in regions in Europe, using Spearman correlation, Poisson regression and artificial neural network methods. Authors show that increase of air pollution including PM2.5 and ozone is associated with the COVID-19 cases in provincial levels in Italy, but found only limited capacity to classify the epidemic escalation in France, Germany and Spain regions. This research supports the hypothesis that 1) atmospheric PM2.5 promotes the formation of persisting forms of airborne droplets with SARS-Cov-2 virus, and thus increase COVID-19 cases, and 2) atmospheric zone reduce the activity of the virus and thus reduce the COVID-19 cases.

Overall, it is a very interesting research, adding evidence of understanding of the spread of on-going COVID-19 pandemic, especially about the environmental/atmospheric pollutions. I would recommend the publication of this manuscript if following comments are addressed.

Major comments:

  1. I have some concerns about missing some variables such as population density of each region and other time-varying atmospheric conditions in author’s analysis. The proposed association could be driven by population density associated with air pollution level and also associated with COVID-19 transmission. I encourage authors to conduct some additional analysis or provide justifications on omitting this variable. In addition, I would recommend authors to add other atmospheric variables such as ambient temperature, wind speed, pressure as supplementary analysis. This weather conditions could potentially also influence the SARS-Cov-2 virus transmission in the environment, and also influence air pollution levels.  Authors should rule out the possibility that, for example temperature, drives the COVID-19 cases but temperature is also associated with PM2.5 and ozone.
  2. I would encourage authors provide additional background in and justifications of their hypothesis in the discussion parts, especially:
  • Related study regarding air pollution and COVID-19 cases/death. One example could be a similar study in United States, even though still pre-print (https://projects.iq.harvard.edu/covid-pm)
  • Related studies on detection of SARS-Cov-2 virus in atmospheric aerosol in Italy. Also just pre-print, but worth noting: (https://www.medrxiv.org/content/10.1101/2020.04.15.20065995v2)

Minor comments:

Line 22-23. In the abstract authors mentioned PM2.5 and ozone are oppositely related with SARS-CoV-2 infection, but the article shows PM2.5 is positively related with SARS-CoV-2 infection. Authors need to make more clear conclusion in the abstract and make it more suitable for medical/public health readers.

Why only restrict analysis before March 25? If possible, I would encourage authors to extend the analysis to a closer date since there are more cases in April and give the analysis more power.

In line 56-57, the definition of PM2.5 and PM10, I would recommend authors to add word ‘aerodynamic’ before word ‘particulate’.

In line 102: what is the meteorological factor value, do you mean the atmospheric air pollution including PM2.5, PM10, NH3 and O3?

Figure 1. I wonder what is the plot look like for France, Germany and Spain? I would like authors to include these other regions in the plot

Figure 2. What is the red line represent, is that LOESS line?

I would recommend author to add a map of the study region (Italy as well as France, Germany and Spain) in the main text so that can give a visual illustration of the study domain. Perhaps authors can move figure 3 in main text to supplement.

Round 2

Reviewer 1 Report

The authors have adequately addressed all my concerns in the revised draft, and I have no further comments.

Author Response

We wish to thank the Reviewer for a prompt reply, and all valuable suggestions and comments that helped us to improve our manuscript.

Reviewer 2 Report

Thanks for the revised version and your reply to my previous comments. I just have a few further questions/comments:

  1. In abstract line 9, please adjust the global total confirmed cases (over two million) to most recent numbers.
  2. I found abstract is a little too lengthly with too much background. Such as the sentence 'The majority...Wuhan." in line 10 -12 can be moved to introduction.
  3. In Figure 3E and Figure 3F, why the predicted and actual cases are order of magnitude different? Figure 3E is the number of cases per day or per season? Similarly, what is the time frame of number of cases in figure 3F?

Congratulate on completing this work. Thanks.
